# Therapeutic downregulation of *neuronal PAS domain 2 (Npas2)* promotes surgical skin wound healing

**Yoichiro Shibuya**[1,2,3†], **Akishige Hokugo**[1,2*†], **Hiroko Okawa**[2,4], **Takeru Kondo**[2,4], **Daniel Khalil**[1], **Lixin Wang**[1], **Yvonne Roca**[1], **Adam Clements**[1], **Hodaka Sasaki**[2], **Ella Berry**[1], **Ichiro Nishimura**[2*], **Reza Jarrahy**[1*]

[1]Regenerative Bioengineering and Repair Laboratory, Division of Plastic and Reconstructive Surgery, Department of Surgery, David Geffen School of Medicine, Los Angeles, United States; [2]Weintraub Center for Reconstructive Biotechnology, Los Angeles, United States; [3]Department of Plastic and Reconstructive Surgery, Faculty of Medicine, University of Tsukuba, Tsukuba, Japan; [4]Division of Molecular and Regenerative Prosthodontics, Tohoku University Graduate School of Dentistry, Miyagi, Japan

**\*For correspondence:**
ahokugo@mednet.ucla.edu (AH);
inishimura@dentistry.ucla.edu (IN);
rjarrahy@mednet.ucla.edu (RJ)

†These authors contributed equally to this work

**Competing interest:** The authors declare that no competing interests exist.

**Abstract** Attempts to minimize scarring remain among the most difficult challenges facing surgeons, despite the use of optimal wound closure techniques. Previously, we reported improved healing of dermal excisional wounds in circadian clock neuronal PAS domain 2 (*Npas2*)-null mice. In this study, we performed high-throughput drug screening to identify a compound that downregulates *Npas2* activity. The hit compound (Dwn1) suppressed circadian *Npas2* expression, increased murine dermal fibroblast cell migration, and decreased collagen synthesis in vitro. Based on the in vitro results, Dwn1 was topically applied to iatrogenic full-thickness dorsal cutaneous wounds in a murine model. The Dwn1-treated dermal wounds healed faster with favorable mechanical strength and developed less granulation tissue than the controls. The expression of type I collagen, Tgfβ1, and α-smooth muscle actin was significantly decreased in Dwn1-treated wounds, suggesting that hypertrophic scarring and myofibroblast differentiation are attenuated by Dwn1 treatment. NPAS2 may represent an important target for therapeutic approaches to optimal surgical wound management.

## Editor's evaluation

This study attempts to use high-throughput drug screening to identify a compound, Dwn1, that downregulates Npas2 activity, and in doing so, increases murine dermal fibroblast cell migration and decreases collagen synthesis in vitro. This work represents a significant advance towards improving the outcomes of surgical wound healing with translational implications.

## Introduction

Postsurgical hypertrophic scarring is relatively frequent, even with careful surgical care aimed at reducing inflammation, angiogenesis, and fibrogenesis (*Shirakami et al., 2020*). Hypertrophic scarring results from the excessive deposition of collagen extracellular matrix (ECM) during wound healing. Since the 1970s, the scar-free wound healing noted in fetal skin during early gestation (*Burrington, 1971*; *Rowlatt, 1979*) has generated extensive research interest in scarless wound healing. Comparative characterizations of fetal and adult wounds have resulted in the identification of numerous

soluble growth factors (*Lichtman et al., 2016*), insoluble ECM proteins (*Buchanan et al., 2009*), and mechanics of tissue contraction (*Parekh and Hebda, 2017*). Research in each of these target areas of interest has generated considerable progress in understanding wound healing. As a result, therapies using individual modulators of the wound healing process, such as transforming growth factor β1 (TGFβ3) (*McCollum et al., 2011*) and IL-10 (*Kieran et al., 2014*), have been studied in clinical trials. To date, however, the outcomes have been mixed, and therefore, the search for new clinically viable solutions to address the problem of pathological wound healing continues.

During skin wound healing, dermal fibroblasts migrate along the ECM scaffold found at the wound edges and into the wound bed (*Tracy et al., 2016*). Mathematical modeling (*McDougall et al., 2006*) and on-chip wound healing assays (*Shabestani Monfared et al., 2020*) suggest that the activation of dermal fibroblast migration plays a critical role in wound closure and the degree of scarring. It has recently been reported that human burn wounds sustained during daylight hours were found to heal faster than those that occurred at night (*Hoyle et al., 2017*). This observation is consistent with observations in circadian entrained/rhythmic hamsters in which cutaneous wounds created 3 hr after light onset healed faster than those created 2 hr after dark onset (*Cable et al., 2017*). Investigation into the mechanisms behind these findings revealed that fibroblast migration behavior was affected by the circadian rhythm (*Hoyle et al., 2017*).

Circadian rhythm provides temporal regulation and coordination of physiological processes and is responsible for functions related to homeostasis (*Franzoni et al., 2017*). The core circadian clock is rigidly maintained by the suprachiasmatic nuclei (SCN) in the hypothalamus (*Akhtar et al., 2002*; *O'Neil et al., 2013*). Clock transcription factors such as circadian locomotor output cycles kaput (CLOCK), neuronal PAS domain 2 (NPAS2), and aryl hydrocarbon receptor nuclear translocator-like (ARNTL, BMAL1) induce the expression of the period (*PER*) and cryptochrome (*CRY*) genes. Heterodimers of PER and CRY molecules, in turn, inhibit the transcriptional activity of *CLOCK*, *NPAS2*, and *BMAL1* (*Takahashi, 2017*). Given the emerging discovery of functional circadian rhythms in all cells and organs (*Matsui et al., 2016*) and the finding that circadian dysregulation is associated with a wide range of diseases (*Miller et al., 2004*; *Sipahi et al., 2014*), clock genes and the products of their expression have become targets in the growing research and clinical field of chronotherapy (*Ye et al., 2018*; *Wei et al., 2018*).

Recently, we discovered that the suppression of *Npas2* via a knockout (KO) mutation in mice significantly accelerated wound healing and skin closure (*Sasaki et al., 2020*). *Npas2* KO dermal fibroblasts exhibited increased cell migration and contraction in vitro (*Sasaki et al., 2020*). While the role of *Npas2* in the central circadian rhythm within the SCN is still unclear, *Npas2* has been identified in peripheral tissues, possibly as a modulator of peripheral circadian processes (*Zhou et al., 1997*; *McNamara et al., 2001*; *Gilles-GonzalezGonzalezGonzalez, 2004*; *Yamamoto et al., 2004*; *Bertolucci et al., 2008*). Microarray analysis of human skin has also identified *Npas2* as one of the upregulated genes associated with aging (*Glass et al., 2013*).

In this study, we hypothesized that the therapeutic suppression of *Npas2* potentiates dermal wound healing with attenuated excessive collagen deposition. Through a high-throughput screening (HTS) process, we identified a small molecule compound that downregulates *Npas2* expression in dermal fibroblasts and results in the accelerated healing of dorsal incisional wounds in mice with minimum scarring.

## Results

### Murine dorsal incisional wound model

While the goal of closing a wound surgically with sutures or other material is to facilitate primary wound healing and yield the best functional and cosmetic outcome, unpredictable factors such as wound infection, dehiscence, or 'spitting' of sutures (*Kim et al., 2018*) can disrupt wound integrity and lead to healing by secondary intention. This process is characterized by the deposition of an excess of granulation tissue and leads to hypertrophic scarring (*Azmat and Council, 2020*). To evaluate the healing of a full-thickness dermal wound by secondary intention, we developed a modification of a previously described murine model (*Ansell et al., 2014*). Two parallel incisions were made on the dorsal skin of each experimental subject animal, and full-thickness dermis was excised to achieve bilateral defects with uniform dimensions. One suture was placed at the midpoint of each

wound to approximate the wound edges, while the anterior and posterior margins of the wound remained separated. This model generated two zones in each incisional wound: the center zone, with suture-supported tissue approximation, and the open peripheral zones (*Figure 1a*). We used visual analog scale (VAS) scoring (*Duncan et al., 2006*) to validate the model. Each wound was evaluated on a daily basis using photographs of the surgical sites (*Figure 1b*). VAS values from the left- and right-side wounds remained comparable between sides, remaining at the moderate healing level until day 6, primarily due to the sustained open wound in the peripheral zones (*Figure 1c*). Hematoxylin-eosin (HE) and Masson's trichrome (MT) staining showed divergent healing patterns in the central wound zones compared to the peripheral zones. The center zones showed good approximation of the wound edges, indicating successful healing by primary intention. In contrast, the peripheral zones formed larger beds of granulation tissue and demonstrated darker MT-stained collagen fibers in the dermis of the wound edges (*Figure 1d*). Quantitative scar index analysis (*Zheng et al., 2011*) showed significantly higher values in the peripheral zones than in the central zones (*Figure 1e*). Additionally, segmented color analysis of MT-stained sections (*Figure 1—figure supplement 1*) revealed that collagen fiber density was significantly higher in the peripheral zones than in the central zones (*Figure 1f*). The central zone wound underwent primary intention healing, while the peripheral zone of this model demonstrated secondary intention healing. Based on these observations, we are confident that the peripheral zone in our model is representative of dermal and subcutaneous wound healing and not the morpho-functional influences of deeper muscle layers that have been described in murine models (*Zomer and Trentin, 2018*). Our full-thickness incisional wound was limited to the *panniculus carnosus* (*Figure 1d*) to minimize the involvement of the muscle layers.

## Identification of therapeutic compounds by high-throughput drug screening for *Npas2* downregulation in dermal Fibroblasts

An HTS system was designed to identify compounds that modulate *Npas2* expression using murine dermal fibroblasts engineered to carry the reporter gene *LacZ* in the *Npas2* allele. We selected small molecule compounds from an FDA-approved drug library (1120 compounds). Reporter gene assays are widely used in HTS to identify compounds that modulate target gene expression (*Siebring-van Olst and van Beusechem, 2018*). In this study, we sought to identify small chemical compounds that downregulate *Npas2* expression. However, we found that some hit compounds that suppressed *Npas2* were false positives due to cytotoxicity that led to cell death or growth suppression.

To mitigate this problem, we performed a separate HTS using the same compound library to evaluate fibroblast migration. Dermal fibroblast migration is an important function that is highly relevant to wound healing (*Liang et al., 2007*). The cell migration HTS was designed with a commercially available 384-well plate with hydrogels printed in the center of each well. The hydrogels were degraded to leave cell-free areas in each well, and fibroblasts migrated at different rates based upon the various influences of the applied small molecule compounds. After the incubation period, fibroblasts were fluorescently stained with calcein AM for the cytoplasm and Hoechst for the nuclei and imaged by Micro Confocal High-Content Imaging System (ImageXpress, Molecular Devices). We developed an algorithm to capture the number and morphology of fibroblasts within the cell-free zone created by the hydrogel.

The HTS data sets from both assays were subjected to in silico computation. *Npas2* expression (Z score $\leq$–2.5) and cell migration (Z score $\geq$2.5) were co-analyzed, resulting in one 'hit' compound that demonstrated an optimum effect on fibroblasts according to these specific metrics, that is, suppression of gene expression and increased migration (*Figure 2a*). Based on these results, the hit compound 'Dwn1' was identified as the candidate compound for use in subsequent experiments.

## Dwn1 downregulates murine dermal fibroblast *Npas2* expression and increases cell migration in vitro

Based on our HTS results, the hit compound Dwn1 was used to study *Npas2* circadian expression and fibroblast migration. The gene expression of *Npas2* in murine dermal fibroblasts was evaluated every 6 hr for 48 hr following synchronization. *Npas2* expression was suppressed by Dwn1 (two-way ANOVA: treatment p = 0.0346) (*Figure 2b*). However, the circadian pattern of *Npas2* expression was not completely diminished (two-way ANOVA: treatment × time p = 0.2664).

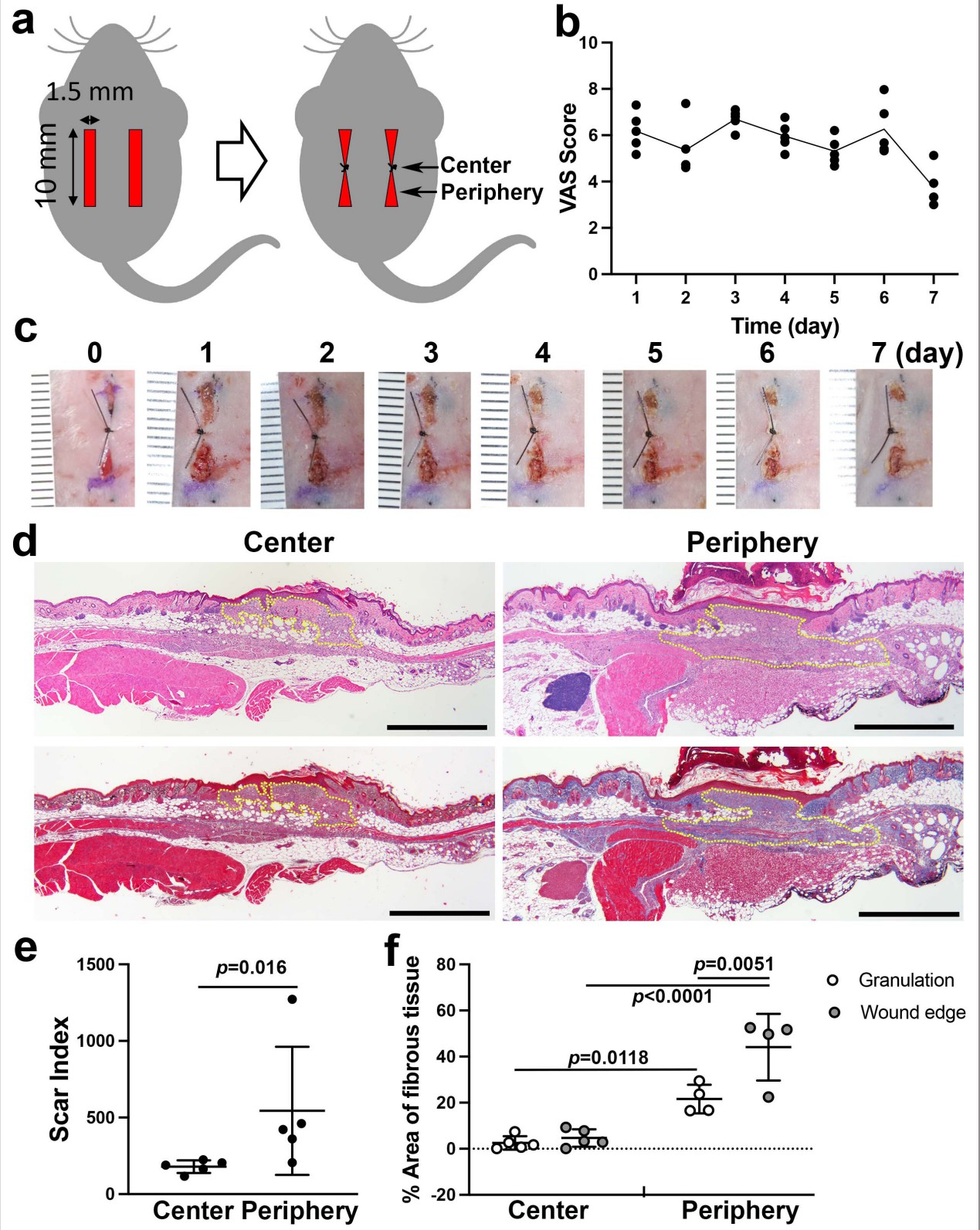

**Figure 1.** Linear wound/scar model of murine dorsal skin. (**a**) Schematic of the animal model. Vertical wounds (10 × 1.5 mm²) on both the left and right sides were made with a double-bladed scalpel. One 5–0 nylon suture was placed at the center of the wound. (**b**) The visual analog scale (VAS) was scored every postoperative day until postoperative day 7 using gross images of the wounds (n = 5 per group). (**c**) Postoperative gross images of the wounds/scars with a ruler in units of mm. (**d**) Histological images of the center (left) and periphery (right) of wounds/scars on postoperative day 7. The

*Figure 1 continued on next page*

Figure 1 continued

upper two were stained with hematoxylin-eosin (HE). The lower two were stained with Masson's trichrome (MT). Yellow dotted lines indicate granulation tissue. Scale bar is 1000 μm. (**e**) The scar index was evaluated using HE-stained slices, and a significantly higher scar index was obtained in peripheral sections than in center sections (n = 5 per group). (**f**) The percent area of fibrous tissue in the peripheral section was significantly higher than that in the central section (n = 5 for center sections, n = 4 for peripheral sections).

The online version of this article includes the following source data and figure supplement(s) for figure 1:

**Source data 1.** Methods for evaluating the scar index and percent area of fibrous tissue.

**Figure supplement 1.** Methods for evaluating the scar index and percent area of fibrous tissue.

The effect of Dwn1 on the migration ability of murine dermal fibroblasts was evaluated by an in vitro wound scratch assay. Dwn1 significantly increased fibroblast migration to the scratched area (two-way ANOVA: treatment × time p = 0.0045) (*Figure 2c*).

To test if the effect of Dwn1 required the *Npas2* axis, we performed in vitro wound scratch assay using dermal fibroblasts derived from *Npas2* KO mice (*Figure 2d*). The cell migration to the scratched area by *Npas2* KO fibroblasts was faster than wild-type (WT) control fibroblasts but was not affected by Dwn1, suggesting that the Dwn1-induced *Npas2* downregulation primarily influenced the increased fibroblast migration. Taken together, the therapeutic hit compound Dwn1 was validated for further evaluation in skin wound healing.

## Dwn1 attenuated fibroblast collagen deposition in vitro

Hypertrophic scarring is the clinical manifestation of a dermal fibroproliferative disorder characterized by excessive collagen deposition by fibroblasts. Normal skin and hypertrophic scars both contain collagen fibers composed of type I and type III collagen molecules; however, hypertrophic scars exhibit irregular organization of collagen bundles (*Cuttle et al., 2005*). Tissue-specific collagen fiber organization is regulated in part by the fibril-associated collagen with interrupted triple helices (FACIT) minor collagen species. Type XIV collagen is a FACIT molecule found in skin and plays an important role in functional dermal collagen fibrogenesis (*Castagnola et al., 1992*; *Berthod et al., 1997*). In this study, collagen deposition and collagen gene expression were evaluated in murine dermal fibroblasts treated with Dwn1. Ascorbic acid-mediated in vitro collagen synthesis was measured by picrosirius red staining. The treatment of dermal fibroblast cultures with Dwn1 significantly decreased collagen deposition (*Figure 3a*).

The mechanism of the reduction in collagen deposition by Dwn1 was investigated by studying the expression of type I, III, and XIV collagen genes. The expression of *Col1a1*, *Col1a2*, and *Col3a1* was affected by Dwn1 supplementation (10 μM) (*Figure 3b*). Type I collagen is composed of two alpha-1 chains and one alpha-2 chain (*Prockop and Kivirikko, 1995*; *Lu et al., 2019*). Although *Col1a1* gene expression was increased on day 7, a reduction in *Col1a2* expression would predict a decrease in the formation of type I collagen heterotrimer. Type III collagen is a homotrimer of alpha-1 chains and occupies the center of type I collagen fibrils (*Keene et al., 1987*). We observed a decrease in *Col3a1* expression in cultures treated with Dwn1, which may decrease overall collagen fiber synthesis. In contrast, the expression of *Col14a1* was significantly increased by Dwn1 in a dose-dependent fashion (*Figure 3b*). Knowing that a lack of type XIV collagen contributes to the abnormally thick type I collagen fibrils in skin in null mutation models (*Ansorge et al., 2009*), the increased FACIT type XIV collagen synthesis induced by exposure to Dwn1 may limit collagen fiber thickness and normalize dermal collagen fiber organization.

## Dwn1 accelerates murine dorsal incisional wound healing with minimal scarring

Murine dorsal incisional wounds were treated topically with vehicle (10% DMSO) or Dwn1 (30 μM dissolved in 10% DMSO) once a day throughout the observation period. One wound was excluded due to postoperative inflammation. Serial photographs depict wound closure at the peripheral zones of the incisional wounds in the Dwn1-treated group (*Figure 4a*). During the first 7 days, VAS scoring yielded scores at the moderate healing level (*Figure 4b*), similar to untreated controls without any significant difference (*Figure 1b*). By comparison, the VAS scores of the Dwn1-treated group gradually decreased, and the VAS time course profile was significantly different from that of the vehicle control

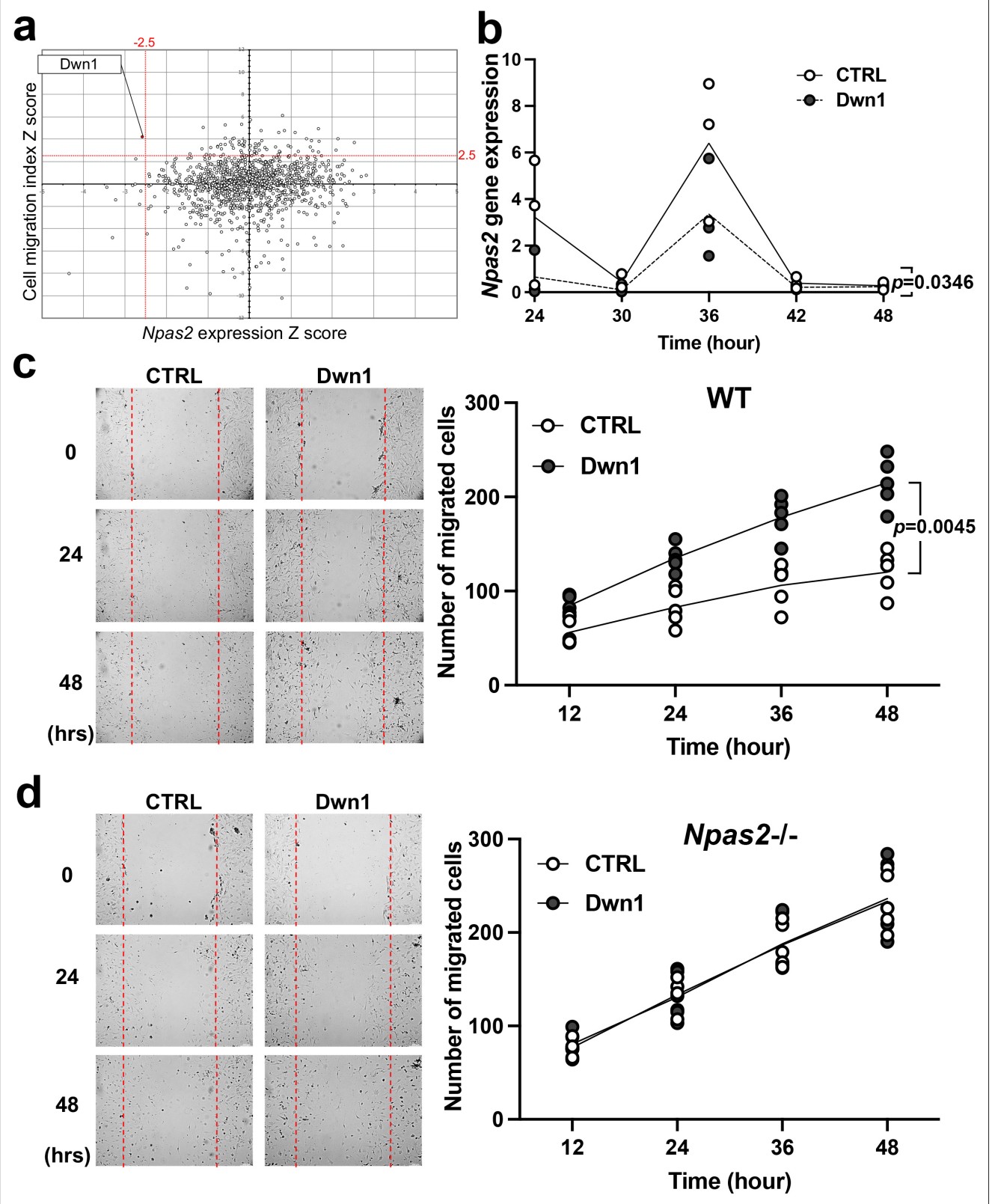

**Figure 2.** Selection and evaluation of the candidate compound Dwn1 for *Npas2* suppression in dermal fibroblasts in vitro. (**a**) A scatter plot of the high-throughput drug screening assay in vitro using the FDA-approved compound library at Molecular Screening Shared Resource (MSSR) at University of California Los Angeles (UCLA). A high absolute value of a negative *Npas2* Z score indicates that *Npas2* expression was highly downregulated (X axis). A high cell viability Z score indicates that fibroblasts had high viability (Y axis). A candidate compound (Dwn1) was selected based on the highest absolute

*Figure 2 continued on next page*

Figure 2 continued

value of the negative product of the *Npas2* Z score and highest viability. (**b**) Circadian *Npas2* expression in murine dermal fibroblasts treated with or without Dwn1 was evaluated (n = 3). The p value (0.0346) in the graph represents the subject of treatment. (**c**) The cell migration of murine dermal fibroblasts treated with Dwn1 was evaluated. The number of cells that migrated toward the central area was counted (n = 5). The p value (0.0045) in the graph represents the subject of treatment and time. (**d**) The cell migration of *Npas2* knockout (KO) fibroblasts treated with Dwn1 was evaluated. The cell migration to the scratched area by *Npas2* KO fibroblasts was not affected by Dwn1.

The online version of this article includes the following source data for figure 2:

**Source data 1.** Selection and evaluation of the candidate compound Dwn1 for Npas2 suppression in dermal fibroblasts in vitro.

(two-way ANOVA: treatment p = 0.0075). On day 14, there was no significant difference in VAS scores between the vehicle control and the Dwn1-treated group (t-test: p = 0.6889) (*Figure 4b*).

Histological cross-sections of the peripheral incisional wound zones demonstrated that smaller amounts of granulation tissue were formed in the Dwn1-treated group than in the vehicle group (*Figure 4c*). The scar index of Dwn1-treated wounds was significantly lower than that of vehicle-treated wounds (*Figure 4d*), indicating that granulation tissue formation was downregulated by Dwn1 treatment. Collagen fiber density within the granulation tissue bed and the wound edge dermis was also lower in Dwn1-treated wounds than in vehicle-treated wounds. The difference in the granulation tissue reached statistical significance (*Figure 4e*).

We have devised a mechanical tensile strength test of murine dorsal incisional wounds. The vehicle control dermal wound tissue and the Dwn1-treated dermal wound tissue were harvested 7 and 14 days of healing and prepared as a standardized dermal strip including the incisional wound in the center. The maximum load (N) to tear the dermal strip was measured as the tensile strength of the wounds (*Figure 4f*, left). The vehicle-treated dermal wound showed minimal tensile strength at 7 days after wounding, which was increased in 14 days after wounding. By contrast, the tensile strength of Dwn1-treated dermal wound was significantly larger than vehicle control dermal wound on both 7 and 14 days after wounding. On day 14, the tensile strength of Dwn1-treated dermal wound nearly reached that of intact skin (*Figure 4f*, right). The vehicle control dermal wound was further examined for 21 days after wounding, which showed significantly less tensile strength than that of intact skin (t-test, p = 0.0048) (*Figure 4—figure supplement 1*). The tensile strength in the vehicle control dermal wound (2.61 ± 0.91) was not significantly increased from that on day 14 (2.04 ± 0.48) (t-test, p = 0.1418), suggesting that this murine dorsal incisional wound model reaches healing 14 days postoperatively, after which recovery of mechanical properties equivalent to intact skin is not expected.

## Dwn1 attenuates collagen gene expression and myofibroblast-related gene expression in the wound edge dermis

To examine lesion-specific gene expression within the incisional wound, granulation tissue (G) and wound edge dermal tissue (W) were separately isolated by laser capture microdissection (LCM) (*Figure 5a*). The isolated RNA was evaluated for the steady-state expression of collagen genes (*Col1a1*, *Col1a2*, *Col3a1*, and *Col14a1*). There was no significant difference in the gene expression of any of these collagen genes between the granulation tissue of Dwn1-treated and vehicle-treated wounds, although Dwn1 treatment minimized the size of granulation tissue, as noted above. The expression of *Col1a1*, *Col1a2*, and *Col3a1* was suppressed in the wound edge dermis, while *Col14a1* gene expression was maintained by Dwn1 treatment (*Figure 5b*).

We also evaluated the expression of the myofibroblast-related genes *Tgfb1* and smooth muscle α2 actin (α-SMA/*Acta2*). Dwn1 treatment significantly decreased the expression of *Tgfb1* and *Acta2* in the wound edge dermis, whereas no effect occurred in the granulation tissue. Immunohistochemical staining demonstrated abundant α-SMA-positive fibroblasts in the wound edge dermis of the vehicle control group. In contrast, Dwn1 treatment virtually eliminated α-SMA-positive fibroblasts (*Figure 5c*), suggesting that Dwn1 prevents myofibroblast differentiation. Collectively, the data supported the therapeutic effect of Dwn1 in the prevention of hypertrophic scar formation.

## Discussion

Scarring is often problematic for patients, especially on exposed parts of the body, such as the craniofacial region or extremities. Surgical approximation of open wound tissues has been traditionally

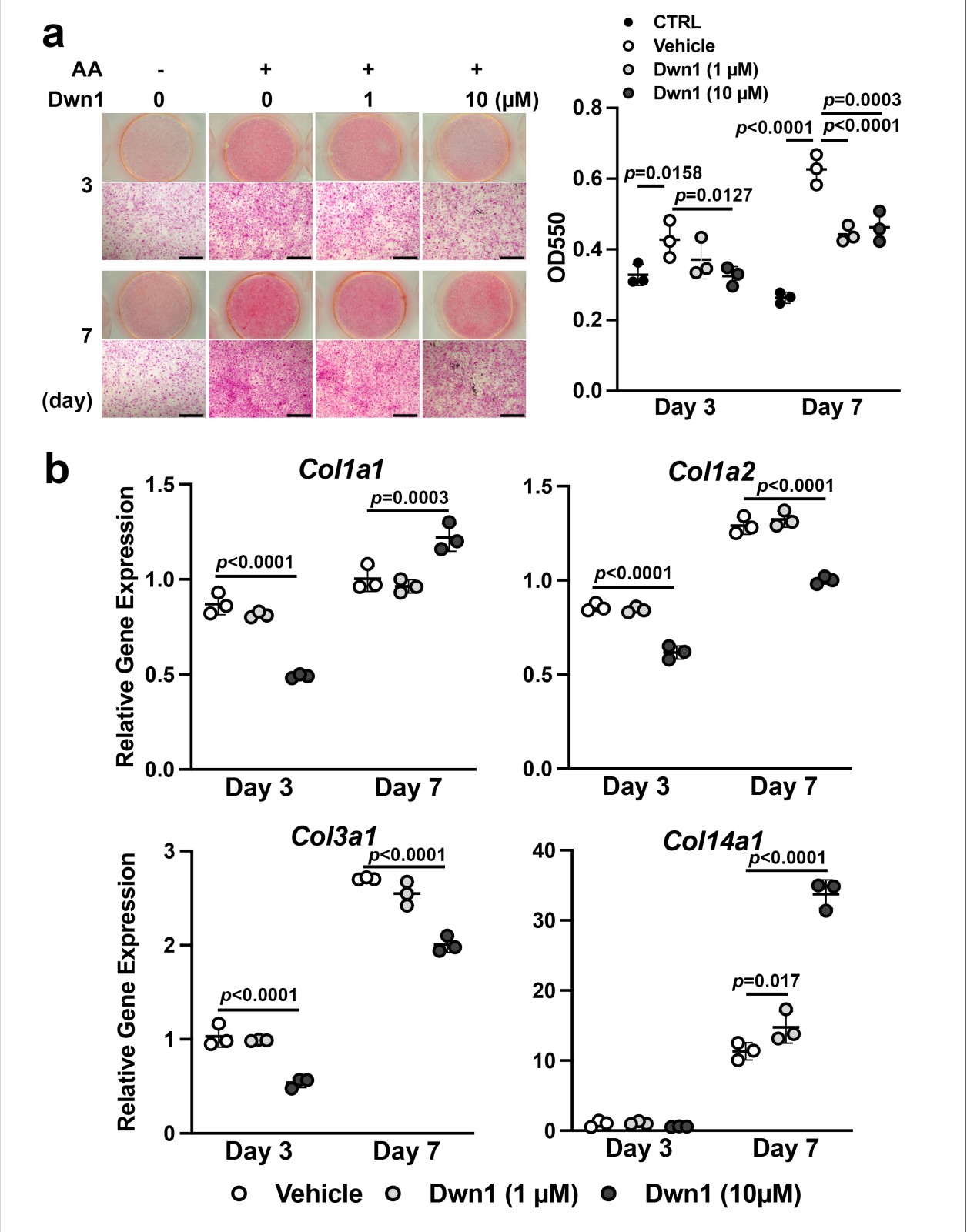

**Figure 3.** Effects of Dwn1 on collagen synthesis in murine dermal fibroblasts in vitro. (**a**) Gross and microscopic images of picrosirius red staining of murine dermal fibroblasts treated with various doses of Dwn1 on days 3 and 7. The right graph represents a quantitative measurement of picrosirius red. (n = 3) AA: L-ascorbic acid. OD: optical density. CTRL: fibroblasts treated with control medium without AA. (**b**) Gene expression of collagen type Iα1 (*Col1a1*), Iα2 (*Col1a2*), IIIα1 (*Col3a1*), and XIV (*Col14a1*) in fibroblasts treated with various doses of Dwn1 on days 3 and 7 (n = 3).

*Figure 3 continued on next page*

*Figure 3 continued*

The online version of this article includes the following source data for figure 3:

**Source data 1.** Effects of Dwn1 on collagen synthesis in murine dermal fibroblasts in vitro.

performed to minimize scarring (*Welshhans and Hom, 2017*). Currently, surgical skin incisions are routinely closed by suturing; however, the surgical closure of a dermal wound does not guarantee adequate approximation of the entire length of the wound. For example, cleft lip scars develop after surgical closure regardless of the corrective techniques used (*Bartkowska and Komisarek, 2020*) or the type or severity of the cleft lip (*Marston et al., 2019*). The prevalence of problematic scar formation has been reported to be between 8% and 47% in pediatric cleft lip patients (*Marston et al., 2019*; *Soltani et al., 2012*). Even in simple facial laceration repair, surgical wounds are not always optimally closed (*Lee et al., 2015*), and small full-thickness dermal gaps in surgical wounds may lead to problems with both form and function. The resultant scarring can lead to numerous psychosocial consequences for patients (*Tebble et al., 2004*), resulting in decreased satisfaction with life, an altered perception of body image, and higher rates of posttraumatic stress disorder, alcoholism, imprisonment, unemployment, or marital discord (*Levine et al., 2005*). There is an acute need to improve surgical wound care.

The surgical incisional wound with the placement of suture heals by the primary closure, or first intension wound healing. However, the surgical incisional wound may result in premature dehiscence or hypertrophic scarring. *Ansell et al., 2014* compared the time course healing of 10 mm long incisional wound and 6 mm diameter excisional punch wound in mice and reported that the incisional wound healed less consistently than the excisional wound. While the excisional wound progressively closed, the incisional wound initially enlarged before wound margin approximation. *Zheng et al., 2011* refined the mouse incisional wound model by creating 10 × 3 mm² full-thickness skin wound to ensure the excision of underlying panniculus carnosus muscle. The present study used this refined mouse incisional wound model with an additional central suture, mimicking the wound dehiscence.

The present incisional wound models including ours have certain limitations. After 2 weeks of wounding, the present incisional wound healed either through the primary closure at the sutured area or the secondary closure. Therefore, this incisional wound model did not develop chronic unhealed wound. Skin is the largest barrier tissue protecting the internal homeostasis from environmental insults. Once injured, the wound healing process initiates the recovery of the barrier function. However, the skin wound healing may be disturbed by various factors such as diabetes to develop unhealed chronic wound. To investigate the mechanism and therapeutic intervention of unhealed chronic wound, mouse excisional skin wound models have been utilized. *Sullivan et al., 2004*, reported a chronic wound model of 6 mm diameter excisional punch wound in diabetic mice. Recently, (*Wu and Landen, 2020*), recommended to create 4 mm diameter excisional punch wound in mice as a standard protocol. However, rodent skin excisional wound (*Chen et al., 2015a*) heals essentially by tissue contraction, whereas human skin wound heals by re-epithelialization. To address this issue, *Wang et al., 2013*, proposed that the mouse excisional wound of 5 mm diameter would be splinted to prevent the rodent-specific wound contraction for the investigation of chronic wound healing relevant to humans. The effect of Dwn1 on the chronic wound must be separately investigated using the splinted excisional wound model.

Our study convincingly demonstrated that Dwn1 treatment significantly improved surgical wound healing by secondary intention (*Figure 4*). The open wounds in the peripheral zones of our surgical incisional wound model were quantitatively depicted by the VAS system. Dwn1 treatment resulted in smaller amounts of granulation tissue, as indicated by lower scar index scores, and less collagen ECM density, as measured by color segmentation analysis of MT-stained histological sections. These observations strongly indicate that Dwn1 facilitates surgical incisional wound healing and results in minimal scarring.

The mechanism of hypertrophic scarring has not been fully elucidated. During the early inflammation phase, cytokines and chemokines, including TGFβ1, promote the recruitment of fibroblasts to initiate wound repair. A hallmark of hypertrophic scar formation is the perturbation of collagen production, resulting in disorganized bundles of collagen ECM (*Hinz et al., 2019*). In the present study, Dwn1 supplementation decreased collagen deposition by dermal fibroblasts in vitro (*Figure 3*) and in murine skin incisional wounds in vivo (*Figure 4*). Type I and III collagen genes downregulated

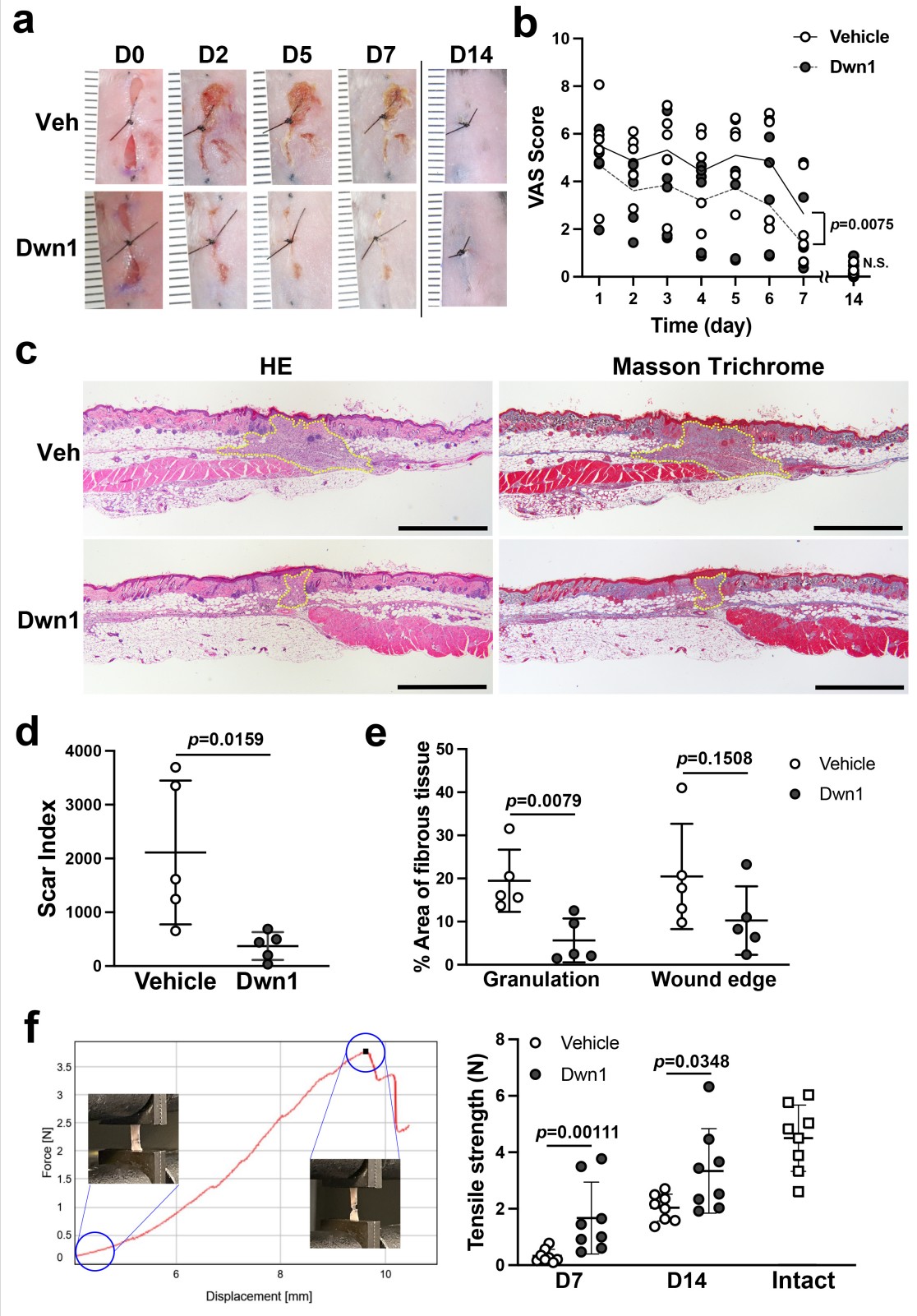

**Figure 4.** Effects of Dwn1 on the murine dorsal incisional wound model. (**a**) Gross images on days 0, 2, 5, 7, and 14 (D0, D2, D5, D7, and D14, respectively) after the surgery and starting topical daily application of the vehicle (10% DMSO) or Dwn1 on the wounds. Veh: vehicle. (**b**) Visual analog scale (VAS) of the wounds treated with vehicle (n = 5) or Dwn1 (n = 6). During the first 7 days, the p value (0.0075) in the graph represents the subject of treatment. There was a significant effect on time (p < 0.0001). The interaction time and treatment had no significant effect (p = 0.3539). On day 14, there

*Figure 4 continued on next page*

*Figure 4 continued*

was no significant difference (p = 0.6889) in VAS scores between the vehicle control (n = 8) and the Dwn1-treated group (n = 4). (**c**) Histological images of peripheral wounds treated with vehicle or Dwn1 on day 7 postoperatively. Yellow dotted lines indicate granulation tissue. Left images were stained with hematoxylin-eosin (HE), and right images were stained with Masson's trichrome (MT). Scale bar is 1000 μm. (**d**) The scar index of wounds treated with vehicle or Dwn1 was evaluated (n = 5). (**e**) The percentage area of fibrous tissue was evaluated using MT-stained slices (n = 5). (**f**) (Left) Representative force-displacement curve of a dermal strip. Inserted images represented the dermal strips at the beginning and termination of the tensile strength test. (Right) The maximum load to tear the dermal strip including the incisional wound at the center treated with vehicle or Dwn1. The tensile strength of Dwn1-treated dermal strips (n = 8) were significantly larger than vehicle-treated dermal strips on both 7 and 14 days after wounding.

The online version of this article includes the following source data and figure supplement(s) for figure 4:

**Source data 1.** The tensile strength in the vehicle control dermal wound on day 21.

**Figure supplement 1.** The tensile strength in the vehicle control dermal wound on day 21.

by Dwn1 in vitro and in vivo may suggest how Dwn1 regulates the attenuation of fibrosis. In addition, type XIV collagen, a member of the FACIT family that has been reported to contribute to normal physiological ECM organization (*Castagnola et al., 1992*; *Berthod et al., 1997*), was upregulated in *Npas2* KO murine fibroblasts in vitro (*Sasaki et al., 2020*). Here, we demonstrated Dwn1 dose-dependent upregulation of the expression of type XIV collagen (*Figure 3b*), suggesting a further mechanism by which Dwn1 contributes to normal ECM organization (*Young et al., 2000*; *Marchant et al., 2002*).

It is widely known that myofibroblasts that secrete excessive collagen into the ECM are involved in the pathogenesis of hypertrophic scars (*Shirakami et al., 2020*; *Tracy et al., 2016*). TGFβ1 acts as a major profibrotic cytokine that induces the differentiation of myofibroblasts and the deposition of excessive collagen ECM (*Darby et al., 2014*). In this study, the myofibroblast markers *Tgfβ1* and α-SMA were significantly decreased in the Dwn1-treated wounded dermis (*Figure 5*). The Dwn1-mediated decrease in myofibroblasts at the wound edge may represent yet another mechanism by which this small compound exerts positive effects on wound healing and prevents hypertrophic scarring.

Our target molecule, NPAS2, is a basic-helix-loop-helix transcription factor dimerized with BMAL1. NPAS2 is a functional ortholog of CLOCK, and in fact, the CLOCK-BMAL1 dimer plays a predominant role in the SCN of the hypothalamus for maintaining the circadian rhythm. *Bmal1* and *Clock* null mutant mice develop severe pathological phenotypes such as early aging (*Kondratov et al., 2006*) and abnormal bone metabolism (*Song et al., 2018*), respectively. Alterations in circadian rhythms increase the risk of developing fibrosis in the liver (*Chen et al., 2010*), lungs (*Dong et al., 2016*), and kidneys (*Chen et al., 2015b*). The role of NPAS2 in skin hypertrophic scar has not been reported. However, *Yang et al., 2019*, reported the upregulation of *NPAS2* in hepatic stellate cells contributing to liver fibrosis. In addition, *Morinaga et al., 2019*, demonstrated that the upregulation of *Npas2* in bone marrow mesenchymal stem cells was induced by the exposure to surface roughened titanium biomaterial resulted in the formation of a thin but dense collagen layer between bone tissue and titanium implant. Thus, we propose that the increased *NPAS2* expression by dermal fibroblasts may contribute to increased collagen deposition potentially leading to hypertrophic scarring.

*Npas2* KO mice demonstrated a relatively limited alteration in circadian behaviors (*Wu et al., 2010*; *Franken et al., 2006*) and liver metabolism (*O'Neil et al., 2013*) and did not exhibit developmental and physiological abnormalities (*Morinaga et al., 2019*). NPAS2-BMAL1 dimers exhibit a high affinity for a *cis*-acting E-box sequence of not only circadian clock genes but also other clock-controlled genes (*Takahashi, 2017*). The mechanism by which the therapeutic suppression of *NPAS2* improves surgical wound healing is currently unknown. Because the genetic and therapeutic suppression of *Npas2* maintained dermal fibroblast-specific type XIV collagen expression and prevented myofibroblast differentiation, we speculate that increased *NPAS2* may be involved in the cellular phenotype alteration leading to abnormal scar formation.

## Conclusion

This study demonstrated that the small molecule compound Dwn1, identified through HTS using *Npas2* as the molecular target, enhanced wound healing in a murine incisional wound model. Dwn1 treatment reduced collagen deposition and accelerated wound closure. Our animal model was designed to mimic a less ideal surgical outcome in incisional wound healing, specifically the development of granulation tissue, which is the precursor to hypertrophic scarring. We identified a possible

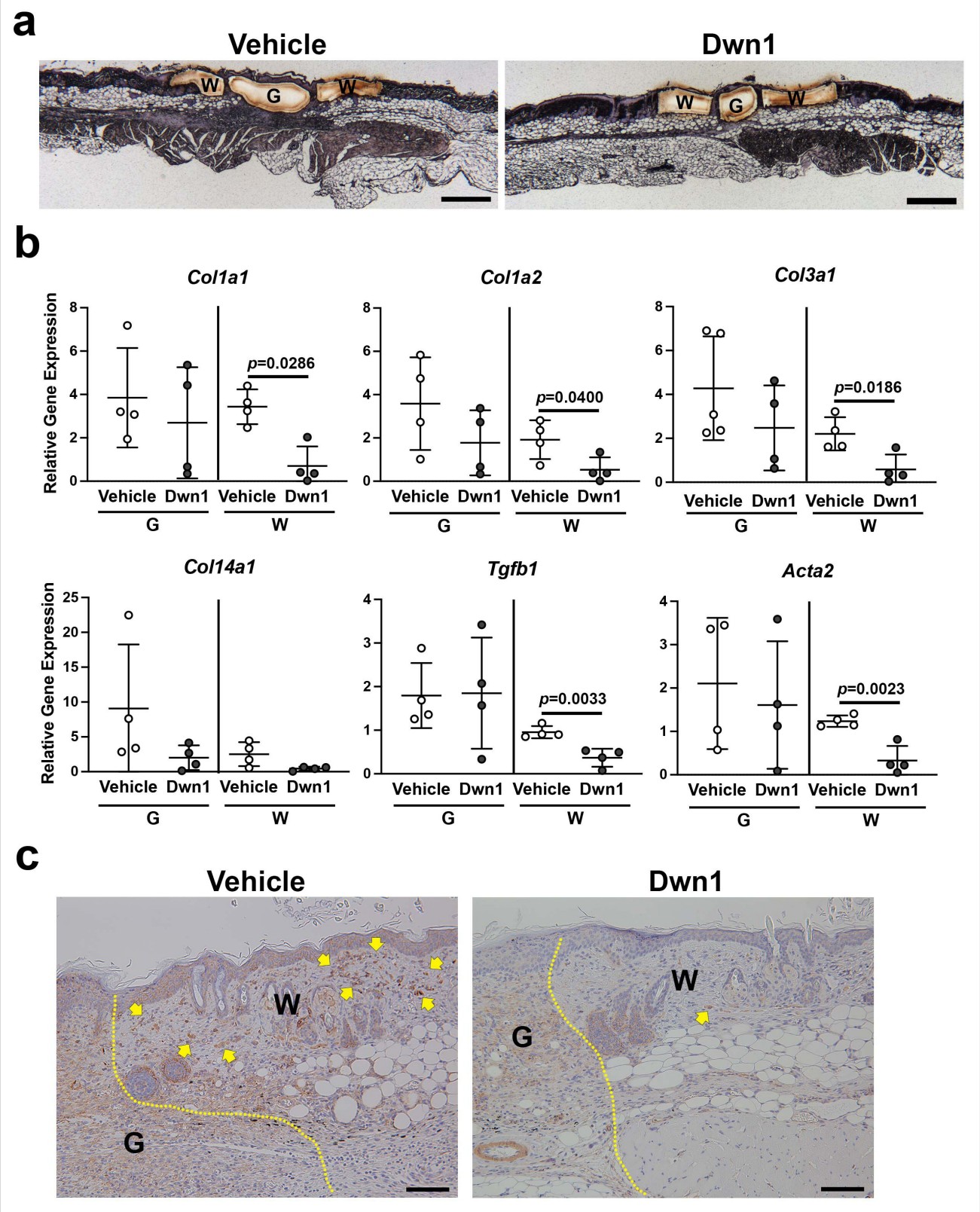

**Figure 5.** Molecular biological effects of Dwn1 on the murine dorsal incisional wound model. (**a**) A typical post-laser capture microdissection (LCM) image. Slides were briefly stained with hematoxylin and eosin before LCM. G: granulation tissue, W: wounded tissue. (**b**) Gene expression of collagen type Iα1 (*Col1a1*), Iα2 (*Col1a2*), IIIα1 (*Col3a1*), XIV (*Col14a1*), *Tgfβ1*, and α-SMA (*Acta2*) in granulation tissue (G) and wounded tissue (W) on day 7 postoperatively. *Gapdh* was used as an internal control. Gene expression of collagen type Iα1, Iα2, IIIα1, Tgfβ1, and α-SMA in the wounded tissue

*Figure 5 continued on next page*

*Figure 5 continued*

treated with Dwn1 was significantly increased versus those treated with the vehicle (n = 5 per group). (**c**) Immunohistochemical staining of α-SMA in wounds treated with vehicle or Dwn1 on postoperative day 7. Yellow dotted lines indicate granulation tissue. Scale bar is 100 µm.

The online version of this article includes the following source data for figure 5:

**Source data 1.** Molecular biological effects of Dwn1 on the murine dorsal incisional wound model.

## Materials and methods

**Key resources table**

| Reagent type (species) or resource | Designation | Source or reference | Identifiers | Additional information |
|---|---|---|---|---|
| Cell line (*Homo sapiens*) | Dermal fibroblast (normal, Adult) | ATCC | CCD-1122Sk | RRID: CVCL_2360 |
| Antibody | Rabbit anti-α-SMA | Abcam | ab32575 | 1/500 |
| Other | Oris Pro Cell Migration Assay 384-well plate | Platypus Technologies | PRO384CMA1 | |
| Other | Masson's trichrome | Polysciences, Inc | 25088–1 | |
| Other | Picrosirius red staining kit | Polysciences, Inc | 24901 | |
| Other | Taqman Gene Expression Assays | Thermo Fisher Scientific | Mm00801666_g1 | *Col1a1* |
| Other | Taqman Gene Expression Assays | Thermo Fisher Scientific | Mm00483888_m1 | *Col1a2* |
| Other | Taqman Gene Expression Assays | Thermo Fisher Scientific | Mm00802300_m1 | *Col3a1* |
| Other | Taqman Gene Expression Assays | Thermo Fisher Scientific | Mm00805269_m1 | *Col14a1* |
| Other | Taqman Gene Expression Assays | Scientific | Mm01178820_m1 | *Tgfb1* |
| Other | Taqman Gene Expression Assays | Scientific | Mm01546133_m1 | *Acta2* |
| Software, algorithm | ImageJ | NIH | http://imagej.nih.gov/ij/ | RRID:SCR_003070 |
| Software, algorithm | GraphPad Prism | GraphPad Software | https://graphpad.com | RRID:SCR_002798 |

### Animal care

All protocols for animal experiments were approved by the University of California Los Angeles (UCLA) Animal Research Committee (ARC# 2003–009) and followed the Public Health Service Policy for the Humane Care and Use of Laboratory Animals and the UCLA Animal Care and Use guidelines. C57Bl/6J WT mice and *Npas2* KO mice (B6.129S6-Npas$^{2tm1Slm}$/J) (Jackson Laboratory) were used in this study. The animals were fed a regular rodent diet and were provided water ad libitum. They were maintained in regular housing conditions with 12 hr light/dark cycles in the Division of Laboratory Animal Medicine at UCLA. The sample size was set based on a previously published manuscript (*Sasaki et al., 2020*).

### Murine dorsal incisional wound healing model

Female WT mice (8–12 weeks of age, approximately 20–25 g) were used. General inhalation anesthesia was obtained using isoflurane (1182097, Henry Schein, Inc) delivered via vaporizer (Somni 19.1, Somni Scientific). The dorsal skin was shaved, and two parallel 10 × 1.5 mm² full-thickness dermal wounds were created using a double-bladed scalpel (S1190D, DoWell Dental Products, Inc). A single 5-0 nylon stitch was placed at the midpoint of each dermal defect, creating partially open wounds that mimicked cutaneous surgical wounds without appropriate wound edge approximation (*Figure 1a*). Photographs and measurements of the wounds were taken every day for 1 week following surgery using a Nikon D40 digital single lens reflex (D-SLR) camera (Nikon, Inc, Tokyo, Japan).

## VAS assessment of incisional wound healing

Gross visual wound assessments were performed by three examiners who were blinded to the type of treatment (control or experimental) each wound received. Sequential postoperative photographs of healing wounds were reviewed by three independent examiners. Scar severity was rated according to a VAS scoring system previously described for use in evaluating human dermal scars.(*Tracy et al., 2016*; *Castagnola et al., 1992*) The examiners were asked to assess the wounds on a scale, with a score of zero indicating a completely healed dermal wound and a score of 10 indicating very poor healing.

## Histological Examination

All animals were euthanized on postoperative day number 7 using carbon dioxide inhalation. For each animal, the entire full-thickness dorsal dermal unit, including both incisional wounds, was harvested and immediately fixed with 10% neutral buffered formalin for 24 hr. Paraffin sections (4 µm) were made perpendicular to the incisional wounds and stained with standard HE or Masson's trichrome (MT). The stained histological sections were examined under a microscope (LMD 7000, Leica Microsystems) equipped with a digital camera (DFC295, Leica Microsystems). Relevant images from the specimens were captured and analyzed using imaging software (Leica LMD software version 8.0.0.6043, Leica Microsystems).

## Scar index analysis

The total scar area of each wound was normalized to the average dermal thickness, as previously described (*Zheng et al., 2011*). The dermal thickness of each dermal section was recorded as the distance from the epidermal-dermal junction to the panniculus carnosus. Four points were used on each sample to calculate the average dermal thickness: two located 200 µm from either wound edge and two located 700 µm from either wound edge. Measurements were taken from the same site on each harvested wound specimen. Scar area was measured on all images. The scar index was calculated as the scar area ($\mu m^2$) divided by the average dermal thickness (µm). (*Figure 1—figure supplement 1a*). Statistical analysis of the measured scar index values was performed using the Mann-Whitney test.

## Quantitative analysis of collagen fiber density on MT-stained slides

Collagen deposition within wounds was quantified on images of MT-stained sections according to a previously reported method with minor modifications (*Kubinova et al., 2017*; *Ying Chen and Xu, 2017*). To evaluate each wound, we randomly selected two images: one from the area of granulation tissue and one from the wounded dermis. These images were obtained with a 40× objective lens using a microscope (Labophot-2, Nikon) equipped with a digital camera (AxioCam, Zeiss) and software (AxioVision Rel. version 4.7, Zeiss). Image analysis was performed using ImageJ (imagej.nih.gov). The split channels function was used to split the original RGB image into red, blue, and green channels. The red channel image was then subtracted from the blue channel image, and a standard threshold range was set for all specimens analyzed. Epithelial and subcutaneous layers and epithelial appendages such as hair follicles and sebaceous glands were eliminated manually. The percentage of collagen density in the tissue area was determined (*Figure 1—figure supplement 1b*). Statistical analysis was performed using the Mann-Whitney test.

## High-throughput drug screening

At the Molecular Screening Shared Resource (MSSR) at UCLA, a drug library of 1120 FDA-approved compounds was screened with two different assays to identify hit compounds with wound healing properties. First, hit compounds involved in the modulation of murine dermal fibroblast *Npas2* expression were identified using HTS. Dermal fibroblasts were isolated from mice engineered to carry the *LacZ* reporter gene in the *Npas2* allele. *LacZ* reporter gene activity has previously been shown to correlate accurately with endogenous *Npas2* expression (*Sasaki et al., 2020*). The cells were cultured in growth medium containing Dulbecco's modified Eagle's medium (DMEM) (11995065, Life Technologies Corp.) with 10% fetal bovine serum (FBS) (1600004, Life Technologies Corp.) and 1% penicillin/streptomycin (15140122, Life Technologies Corp.). Each well in 384-well plates (781906, Greiner Bio-One) was filled with 25 µL of non-phenol red DMEM (31053036, Life Technologies Corp.) containing 10% FBS and 1% PS and 50 nL of FDA-approved compounds (final concentration: 1 µM) using a

pin tool (Biomek FX, Beckman Coulter). Cells were added to each well (1500 cells/25 µL) and incubated at room temperature for 1 hr, followed by incubation at 37°C and 5% carbon dioxide for 48 hr. To measure *Npas2-LacZ* expression, β-galactosidase activity was measured using a Beta-Glo Assay System (E4720, Promega). The *Npas2-LacZ* expression data were uploaded to an online data analysis tool (CDD Vault, Collaborative Drug Discovery Inc), on which data were normalized and the Z-factor was calculated.

Separately, the same compound library was screened for human dermal fibroblast migration. A commercially available human dermal fibroblast cell line (CCD-1122Sk, ATCC) (3000 cells/25 µL) was applied to an Oris Pro Cell Migration Assay 384-well plate (PRO384CMA1, Platypus Technologies), which contains a water-soluble biocompatible gel that creates a center cell-free detection zone for cell migration in each well. After the cells were plated, the plates were centrifuged at 200× *g* for 5 min. After 1 hr of incubation at room temperature for cell attachment, the compounds were added using a 250 nL pin tool and incubated at 37°C in a carbon dioxide incubator. After 48 hr of incubation, 25 µL of staining solution (Calcein-AM and Hoechst, Life Technologies Corp.) was added to each well. After another period of centrifugation at 200× *g* for 5 min, the plates were incubated for 20 min at room temperature, and each well was imaged by the Micro Confocal High-Content Imaging System (ImageXpress, Molecular Devices). The cells that migrated into the detection zone were counted using a customized computer program (CDD Vault, Collaborative Drug Discovery, Burlingame, CA), and the Z-factor was calculated.

### Npas2 gene expression in murine dermal fibroblasts treated with Dwn1

Primary dermal fibroblasts derived from WT mice were harvested as previously reported (*Glass et al., 2013*) and cultured as described above. Fibroblast cultures were treated with 10 µM Dwn1, a hit compound identified via the high-throughput drug screening process, to modulate Npas2 expression and fibroblast migration. After synchronization using 10 nM dexamethasone, total RNA from the fibroblasts cultured with 10 µM Dwn1 was extracted every 6 hr from hours 24 to 48 (RNeasy Plus Mini Kit, Qiagen), followed by cDNA synthesis (SuperScript VILO cDNA Synthesis Kit, Thermo Fisher Scientific). TaqMan-based qRT-PCR was performed using a primer/probe mix, *Npas2* (Mm01239312_m1, Thermo Fisher Scientific), with mouse *Gapdh* endogenous control mix (4352339E, Thermo Fisher Scientific). Statistical analysis was performed by two-way ANOVA.

### In vitro wound healing scratch assay

Murine dermal fibroblasts derived from either WT or *Npas2* KO mice were seeded into a six-well plate with or without 10 µM Dwn1 supplementation. After 2 hr, the cells were scratched with a 20 µL plastic pipette, and the debris was washed out with medium. The scratched regions were imaged every 12 hr by time-lapse photomicrography (LAX S, Leica Microsystems). The number of cells that migrated into the scratched regions was counted at hours 0, 12, 24, 36, and 48. Statistical analysis was performed by two-way ANOVA.

### Collagen synthesis by murine dermal fibroblasts in vitro

Fibroblasts were cultured in growth medium and L-ascorbic acid 2-phosphate (Sigma-Aldrich Corp.) with either 0, 1, or 10 µM Dwn1 supplementation to evaluate collagen synthesis. Control fibroblast cultures were also performed using growth medium only. Picrosirius red staining was performed using a commercially available kit (24901, Polysciences, Inc) at day 3 and day 7, according to the manufacturer's protocol. Gross images were acquired with a Nikon D40 D-SLR camera (Nikon, Inc, Tokyo, Japan). Collagen deposition was quantified by analyzing absorbance of 550 nm with a plate reader (SYNERGY H1, BioTek). Statistical analysis was performed by Dunnett's multiple comparison test at each time point.

### Collagen gene expression by murine dermal fibroblasts in vitro

The gene expression of collagen types I, III, and XIV was determined at day 3 and day 7 of culture. Using isolated RNA, TaqMan-based qRT-PCR was performed with primer/probe sets: *Col1a1* (Mm00801666_g1), *Col1a2* (Mm00483888_m1), *Col3a1* (Mm00802300_m1), and *Col14a1* (Mm00805269_m1). *Gapdh* was used as an internal control. Statistical analysis was performed by Dunnett's multiple comparison test at each time point.

## Evaluation of wound healing by Dwn1 using a murine dorsal incisional wound model

Two parallel full-thickness dorsal incisional wounds were placed on female WT mice (8–12 weeks of age, approximately 20–25 g) as described above. Following surgery, 20 µL of 10% DMSO (Sigma-Aldrich Corp.) was applied every day to one wound as a control, and 20 µL of 30 µM Dwn1 dissolved in 10% DMSO was applied to the other wound. Standardized digital photographs and measurements of all wounds were taken daily postoperatively.

Daily visual wound assessments were performed to evaluate the effects of Dwn1 on wound healing using the VAS scoring system described above. Statistical analysis was performed by two-way ANOVA. In addition, the collagen fiber density in the granulation tissue and the tissue surrounding the wound was measured as described above. Statistical analysis was performed by the Mann-Whitney test.

Animals were euthanized by carbon dioxide inhalation on postoperative day 7, 14, and 21 for the harvesting of dorsal skin. After the samples were harvested, the dorsal skin tissues were cut out to prepare identical dermal strips (15 mm long and 5 mm wide) standardized perpendicularly to the incision with the incisional wound at the center, using a surgical scalpel blade. The dermal strips were then mounted in metal clamps with pneumatic grips on Instron 5564 materials testing machine (Instron, Canton, MA). A load was applied at a constant tension rate of 5 mm/min until the dermal strips failure and force-displacement curves were recorded. The maximum load (N) to tear dermal strips was measured as the tensile strength of the wounds.

LCM of the dermal samples on formalin-fixed paraffin-embedded (FFPE) blocks was performed with an LMD 7000 (Leica Microsystems) according to the manufacturer's protocol, including brief HE staining of slides. RNA extraction was performed with an RNeasy FFPE Kit (Qiagen GmbH) followed by cDNA synthesis. TaqMan-based qRT-PCR was performed to evaluate the gene expression of collagen-related genes as well as *Tgfb1* (Mm01178820_m1) and α-SMA (*Acta2*: Mm01546133_m1). *Gapdh* was used as an internal control. LCM samples contain a small number of cells, and RNA degradation has been reported to contribute to the large variation (*Frost et al., 2001*; *Clément-Ziza et al., 2008*; *Ståhlberg and Kubista, 2014*; *Cai et al., 2014*). In this study, Z scores (*Mowbray et al., 2019*; *Roden et al., 2014*) were utilized to identify outliers that had scores with absolute values of 1.25 or greater for each Ct value in the statistical evaluation of RT-qPCR results. Statistical analysis was performed by the Mann-Whitney test.

To perform immunohistochemistry of α-SMA, histological sections were deparaffinized and rehydrated through graded ethanol. Endogenous peroxidase activity was blocked with 3% hydrogen peroxide in methanol for 10 min. Heat-induced antigen retrieval was carried out for all sections in AR6 buffer (AR6001KT, PerkinElmer), pH = 6.00, using a BioCare Decloaker at 95°C for 25 min. The slides were then stained with rabbit anti-α-SMA (ab32575, Abcam) at 1/500 dilution for 1 hr at room temperature. The signal was detected using the Dako Envision + System Labeled Polymer HRP anti rabbit (K4003, Agilent). All sections were visualized with the diaminobenzidine reaction and counterstained with hematoxylin. Images were acquired with a 2× objective lens from each slide using a microscope (Labophot-2, Nikon) equipped with a digital camera (AxioCam, Zeiss) and AxioVision Rel software, version 4.7 (Zeiss).

## Statistical information

In vitro experiments were performed in biological triplicate. For biological replicates, each experiment was performed with two technical replicates. All raw data are represented in the graphs, unless otherwise specified in each section. Statistical analyses were performed as described above for each separate experiment.

## Acknowledgements

This work was generously supported in part by the Annenberg Fund for Craniofacial Surgery and Research at UCLA (RJ), the Plastic Surgery Foundation (AH), and the UCLA Innovation Fund (IN). We thank Dr Robert Damoiseux, Director of UCLA MSSR, for his consultation for high-throughput drug screening. We also thank the Translational Pathology Core Laboratory (TPCL) in the UCLA Department of Pathology and Laboratory Medicine for histology preparation and immunohistochemistry, and Dr Chase Linsley in the UCLA Bioengineering for Instron test guidance.

## Additional information

### Funding

| Funder | Grant reference number | Author |
|---|---|---|
| Annenberg Foundation | | Reza Jarrahy |
| Plastic Surgery Foundation | | Akishige Hokugo |
| University of California, Los Angeles | Innovation Fund | Ichiro Nishimura |

The funders had no role in study design, data collection and interpretation, or the decision to submit the work for publication.

### Author contributions

Yoichiro Shibuya, Data curation, Formal analysis, Validation, Visualization, Writing – original draft; Akishige Hokugo, Conceptualization, Data curation, Formal analysis, Funding acquisition, Methodology, Project administration, Resources, Supervision, Validation, Visualization, Writing – original draft, Writing – review and editing; Hiroko Okawa, Takeru Kondo, Lixin Wang, Hodaka Sasaki, Data curation, Formal analysis; Daniel Khalil, Data curation, Formal analysis, Writing – original draft; Yvonne Roca, Adam Clements, Formal analysis; Ella Berry, Writing – review and editing; Ichiro Nishimura, Conceptualization, Funding acquisition, Methodology, Resources, Supervision, Writing – review and editing; Reza Jarrahy, Conceptualization, Funding acquisition, Methodology, Project administration, Resources, Supervision, Writing – review and editing

### Author ORCIDs

Yoichiro Shibuya (iD) http://orcid.org/0000-0002-0558-5154
Akishige Hokugo (iD) http://orcid.org/0000-0002-7097-3364
Ichiro Nishimura (iD) http://orcid.org/0000-0002-3749-9445
Reza Jarrahy (iD) http://orcid.org/0000-0003-2518-4697

### Ethics

All protocols for animal experiments were approved by the University of California Los Angeles (UCLA) Animal Research Committee (ARC# 2003-009) and followed the Public Health Service Policy for the Humane Care and Use of Laboratory Animals and the UCLA Animal Care and Use guidelines.

### Decision letter and Author response

Decision letter https://doi.org/10.7554/eLife.71074.sa1
Author response https://doi.org/10.7554/eLife.71074.sa2

## Additional files

### Supplementary files

• Transparent reporting form

### Data availability

Raw data were represented in the graphs.

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
