## [Editor Report]

This study attempts to use high-throughput drug screening to identify a compound, Dwn1, that downregulates Npas2 activity, and in doing so, increases murine dermal fibroblast cell migration and decreases collagen synthesis in vitro. This work represents a significant advance towards improving the outcomes of surgical wound healing with translational implications.

---

## [Decision Letter]

**Decision letter after peer review:**

Thank you for submitting your article "Therapeutic downregulation of neuronal PAS domain 2 (Npas2) promotes surgical skin wound healing" for consideration by *eLife*. Your article has been reviewed by 2 peer reviewers, and the evaluation has been overseen by a Reviewing Editor and Mone Zaidi as the Senior Editor. The following individuals involved in review of your submission have agreed to reveal their identity: Naoki Morimoto (Reviewer #1); Brian S Kim (Reviewer #2).

Essential revisions:

From Reviewer #1:

1) Authors wound model using incisional wounds sutured at the center seemed to be narrow to evaluate the healing process. It will be clear they use full-thickness skin defects more than 6 mm.

2) The observation period of 7 days seems to be short. How about the differences of scaring after two or three weeks?

3) They compared the area of the formed granulation tissue in vivo. I think the mechanical strength of the wound might be compared to evaluate the healing. If the strength of Dwn1 group was not enough compared to the control group, Dwn1 could not be used for wound closure.

From Reviewer #2:

1) Are there any available datasets on human wounds or hypertrophic scars that implicate Npas2?

2) Can more information be provided the specific chemical/biological properties of Dwn1?

3) Dwn1 could have off-target effects. Did the authors confirm that its effects are entirely dependent on Npas2 by using Npas2-/- mice?

---

## [Author Response]

Essential revisions:From Reviewer #1:1) Authors wound model using incisional wounds sutured at the center seemed to be narrow to evaluate the healing process. It will be clear they use full-thickness skin defects more than 6 mm.

We agree that the use of adequate skin wound models is critical for the data interpretation. We revised Discussion to add the contemporary summary of mouse skin wound models and identified the uniqueness and limitations of our “mouse skin incisional model”.

Revised Discussion:

“The surgical incisional wound with the placement of suture heals by the primary closure, or first intension wound healing. […] The effect of Dwn1 on the chronic wound must be separately investigated using the splinted excisional wound model.”

2) The observation period of 7 days seems to be short. How about the differences of scaring after two or three weeks?

We appreciate this suggestion. A new experiment was performed to evaluate the long-term wound healing of our model for 2 weeks. The visual analog scale (VAS) was added to the revised manuscript. The control incisional wound appeared to be healed by 2 weeks after wounding by VAS measurement. We revised Results (Lines 197-207 and Figure 4a and b).

3) They compared the area of the formed granulation tissue in vivo. I think the mechanical strength of the wound might be compared to evaluate the healing. If the strength of Dwn1 group was not enough compared to the control group, Dwn1 could not be used for wound closure.

We also appreciate this suggestion and performed an additional tensile strength experiment. We revised the Results section (Lines 215-229, Figure 4f, and Figure 4-supplement1) and the Materials and methods section (Lines 488-495)

Results:

“We have devised a mechanical tensile strength test of murine dorsal incisional wounds. […] The tensile strength in the vehicle control dermal wound (2.61± 0.91) was not significantly increased from that on day 14 (2.04± 0.48) (T-test, P=0.1418), suggesting that this murine dorsal incisional wound model reaches healing 14 days postoperatively, after which recovery of mechanical properties equivalent to intact skin is not expected.”

From Reviewer #2:1) Are there any available datasets on human wounds or hypertrophic scars that implicate Npas2?

We reviewed literatures and the below section was included in the revision.

Revised Discussion:

“The role of NPAS2 in skin hypertrophic scar has not been reported. However, Yang et al., (65) reported the upregulation of NPAS2 in hepatic stellate cells contributing to liver fibrosis. In addition, Morinaga et al., (66) demonstrated that the upregulation of Npas2 in bone marrow mesenchymal stem cells was induced by the exposure to surface roughened titanium biomaterial resulted in the formation of a thin but dense collagen layer between bone tissue and titanium implant. Thus, we propose that the increased Npas2 expression by dermal fibroblasts may contribute to increased collagen deposition potentially leading to hypertrophic scarring.”

2) Can more information be provided the specific chemical/biological properties of Dwn1?

We have been advised by UCLA administration not to disclose Dwn1 citing its ongoing process of intellectual property protection at this point.

3) Dwn1 could have off-target effects. Did the authors confirm that its effects are entirely dependent on Npas2 by using Npas2-/- mice?

This is an important question. We appreciate this suggestion and have performed an additional in vitro wound scratch assay using dermal fibroblasts derived from Npas2 KO mice. We revised the Results section and Figure 2c and d and the Materials and methods section.

Revised Results:

“The effect of Dwn1 on the migration ability of murine dermal fibroblasts was evaluated by an in vitro wound scratch assay. […] Taken together, the therapeutic hit compound Dwn1 was validated for further evaluation in skin wound healing.”